# Knowledge of Tuberculosis preventive treatment among people living with HIV: A cross-sectional survey in selected regions of Tanzania

Felix Christopher Alexander[1]*, Johnson Jeremia Mshiu[1], Anelisa Martin Rushaigo[1], Erick Josephat Mgina[2], Victor Enock Wiketye[3], Segere Chacha Mtundi[1], Sylvia Thomas Haule[1], Michael Kipenda Katende[1], Esther Manka Evarist[1], Luciana Charles Kapama[1], Aloisia Ibrahim Shemdoe[1], Charles Elias Makasi[1], Majaha Melkisedeck Lolakeha[1], Victor Kenedy Minja[1], Omary Abdallah Kimbute[1], Kunda John Stephen[1], Nyagosya Segere Range[1], Werner Meinrad Maokola[4], Bernard James Ngowi[1], Vitus Alberto Nyigo[1], Andrew Martin Kilale[1]

**1** National Institute for Medical Research-Muhimbili Center, Dar es Salaam, Tanzania, **2** National Institute for Medical Research headquarter, Dar es Salaam, Tanzania, **3** National Institute for Medical Research-Ngongongare Station, Arusha, Tanzania, **4** National AIDS Control Program (NACP) (Now known as National AIDS, STIs and Hepatitis Control Program (NASHCoP), Dodoma, Tanzania

* felixie1986@gmail.com

## Abstract

### Background

Tuberculosis remains a significant global health concern, especially for People Living with HIV, who are at an increased risk of severe TB disease. Despite the availability of TB Preventive Treatment, knowledge gaps persist among People Living with HIV regarding its importance, accessibility, and administration. The study aimed to assess TPT knowledge levels and determinants among People Living with HIV in Tanzania.

### Methods

A cross-sectional survey was conducted from April to May 2023 in 12 regions of mainland Tanzania. The study included PLHIV aged 18 years and above, receiving HIV care in selected Care and Treatment Centers. Data were collected through face-to-face interviews using a semi-structured questionnaire covering sociodemographic characteristics and Tuberculosis preventive treatment knowledge. Descriptive statistics, chi-square tests, and logistic regression analyses were employed for data analysis.

### Results

Out of the 391 People Living with HIV interviewed, 71.4% demonstrated adequate Tuberculosis preventive treatment knowledge. Female participants, those attending urban health facilities, and individuals with longer durations of HIV care exhibited higher Tuberculosis

**Funding:** Our study received funding support from the Global Fund to fight against HIV, TB and Malaria in Tanzania through the NASHCoP. However, the funder did not take part in the study design, data collection and analysis, decision to publish, nor preparation of this manuscript. AMK received the fund. https://www.theglobalfund.org/en/results/.

**Competing interests:** The authors have declared that no competing interests exist

preventive treatment knowledge levels. However, knowledge disparities persisted based on demographic characteristics such as gender and location of health facilities.

## Conclusion

While a considerable portion of People Living with HIV demonstrated adequate higher Tuberculosis preventive treatment knowledge, addressing gaps among those with lower understanding is crucial. Targeted education campaigns tailored to the needs of People Living with HIV, especially in rural areas and among male populations, are essential. Collaborative efforts between national health programs and community organizations are vital to integrate Tuberculosis preventive treatment awareness effectively into comprehensive HIV care programs, ultimately reducing the burden of Tuberculosis among People Living with HIV and the general population.

## Introduction

Tuberculosis is still a public health threat affecting a significant proportion of global populations particularly People Living with HIV (PLHIV) [1]. People living with HIV and children, especially those in the under-five age group are at an increased risk of severe forms of TB disease [2]. In 2022, TB claimed the lives of 1.3 million people worldwide, including 167,000 HIV-positive individuals [1]. As captured in data between the years 2020 through 2022, TB was the second most common infectious killer globally, after COVID-19, surpassing HIV and AIDS [3]. The World Health Organization's 2030 End TB strategy recommends TB Preventive Treatment (TPT) for Individuals who potentially carry a TB infection or have been exposed to the bacterium who are more susceptible to developing TB compared to the general population. TPT is regarded as one of the most important public health interventions to prevent TB in both the community and individuals thus contributing to the broader goal of reducing the global burden of TB [4]

TB remains a significant comorbidity among PLHIV in Tanzania. In 2022, the proportion of people with TB coinfected with HIV stood at 17% [5]. The National TB and Leprosy Program (NTLP) adopted the WHO recommendations for TB prevention among high-risk populations thus recommending Daily Isoniazid as the main preventive drug for PLHIV and under-five household contacts of people with active TB. Being the most widely used TB preventive drug worldwide, with a low likelihood of drug-drug interactions, good tolerance, and much evidence of effectiveness, IPT has been shown to reduce mortality by 50% and more than 70% incidence among these populations in high TB-burden countries [6].

Knowledge of TB preventive treatment is a fundamental aspect of TB prevention among at-risk populations such as PLHIV. Education campaigns and outreach efforts are essential to ensure that accurate information reaches diverse populations, addressing disparities in knowledge and promoting a comprehensive approach to TB prevention. While evidence indicates a significant gap in IPT uptake in Tanzania [7], no studies have explored the knowledge of IPT treatment courses among PLHIV, despite its crucial role in promoting drug uptake. Similarly, rare studies have investigated this area in the Sub-Saharan region where TB prevention efforts are needed most. A study conducted in Southeast Nigeria revealed that about 60.5% of PLHIV had a lower level of IPT knowledge [8]

Tuberculosis-preventive drugs have evolved paving the way to shorter regimens compared to Isoniazid which is given daily for six months [4]. An important benefit of advancements in the pharmaceutical industry is the ability to significantly reduce the duration of TPT with

certain medications such as rifampicin and rifapentine (rifamycin). The Ministry of Health (MoH) Tanzania is expected to roll out a shorter regimen of TB prevention service to extend its efforts in TB Prevention in the country by 2024. According to existing evidence, knowledge of the targeted populations on the treatment cascade is an important factor for its successful delivery and uptake. This study was conducted to assess the level and determinants of knowledge of IPT among PLHIV. The findings of this study will provide valuable insights to the NTLP and other stakeholders in the TB field, highlighting existing gaps and offering potential strategies to enhance awareness of TPT thereby promoting optimal health outcomes among the target populations.

## Materials and methods

### Study design

A health facility-based cross-sectional survey was conducted from April 2023 to May 2023.

### Study area

The study was conducted in 12 regions of mainland Tanzania, namely, Dar es Salaam, Mara, Dodoma, Ruvuma, Tanga, Simiyu, Mbeya, Kagera, Mwanza, Shinyanga, Pwani, and Geita in mainland Tanzania. The regions are supported by the Global Fund to combat TB, HIV, and Malaria.

### Study population

The study enrolled participants aged 18 years and above, who were receiving HIV care and treatment in CTCs in the study regions above.

### Sample size and sampling procedure

The sample size for the study was obtained using a single proportion formula with the assumption that 50% of the PLHIV in the population are knowledgeable about IPT. An intra-class coefficient (ICC) of 0.01, a cluster size of 11, and an expected response rate of 99% meant the study was estimated to require a sample size of 427 for estimating the expected proportion with 5% absolute precision and 95% confidence. Thus, 39 clusters, each of 11 sizes were calculated to be selected for the study, and the list of all health facilities offering HIV care and treatment in each study region was obtained. A probability proportional to the size of PLHIV was used to select 39 health facilities in 12 regions. A systematic sampling procedure was used to select eleven PLHIV from selected facilities for the interview.

### Data collection

Face-to-face interviews were conducted to gather data using a semi-structured questionnaire altered and modified from earlier research [8]. The questionnaire was designed and uploaded to a server located at the National Institute for Medical Research (NIMR), Muhimbili Centre. Later questionnaires were downloaded onto Android mobile devices using the Open Data Kit (ODK) application for data collection. The information collected included participants' social demographic characteristics, duration of care, and knowledge of IPT.

### Measurement of variables

In this study, knowledge regarding IPT was evaluated through a set of four questions, each accompanied by multiple-choice options. These questions aimed to assess participants'

comprehension of several crucial aspects: the preventive effectiveness of IPT against TB, awareness of where IPT services are accessible, understanding of the duration of IPT dose, and recognition of the specific demographic groups targeted for IPT administration. The score set ranged from 0 to 16, with a higher score indicating better knowledge and a lower score indicating poor knowledge. To categorize knowledge scores, a Bloom's cut-off system was applied, following the methodology outlined by [9, 10]. Specifically, a cut-off of $\geq 60\%$ was utilized to classify participants as having adequate knowledge of IPT.

## Data analysis

Stata version 14.2 (Stata Corp, Texas-USA) was used for data management and statistical analysis. Descriptive statistics were employed to characterize participants' sociodemographic profiles and their understanding of IPT. Bloom's cut-off point was used to determine adequate knowledge ($\geq 60\%$). Pearson's chi-square test was utilized to assess the relationship between participants' knowledge of IPT and their sociodemographic attributes. To accommodate potential clustering effects, a generalized linear mixed model was applied. In this model, knowledge served as the dependent variable, while sociodemographic characteristics were used as predictors. Odds Ratios were computed with a 95% confidence interval, and statistical significance was determined at $p < 0.05$.

## Ethical consideration

Ethical clearance was secured from the National Health Research Ethics Committee (NatHREC) of the National Institute for Medical Research, with Certificate No. NIMR/HQ/ R.8a/Vol IX/4256. Additionally, permission to conduct the study was obtained from relevant authorities in the selected regions, districts, and health facilities. Before each interview, participants were briefed on the study's objectives and assured that participation was voluntary. Written consent was obtained from all participants before their involvement in the study.

## Results

### Description of study participants

A total of 391 PLHIV were interviewed. The majority of respondents 279(71.4%) were female, while males accounted for 112 (28.6%). In terms of age group, more than one-third 154 (39.4%) were aged between 32–45 years, and more than quarter 121(30.9%) were aged between 46–59 years. The mean age was 44 years with a standard deviation of 12.7. Regarding marital status, nearly half of the respondents 186 (47.6%) were married, followed by those who were divorced, widowed, or separated 128 (32.7%), and single individuals 77 (19.7%). The educational background varied, with the majority 282 (72.1%) having completed primary education, followed by those with no formal education 59 (15.1%) (**Table 1**).

### Knowledge of IPT among PLHIV

A total of 391 PLHIV were interviewed to assess IPT knowledge. When asked about the effectiveness of IPT in preventing TB, a substantial percentage 188 (48.1%) responded affirmatively, while 121 (31%) expressed absolute confidence in the capacity of IPT to prevent TB disease. However, a notable portion of respondents, 60 (15.4%) were uncertain about IPT's efficacy. In terms of access to IPT services, half of the respondent 196 (50.1%) knew where to access them, with 135 (34.5%) expressing absolute certainty. Regarding the understanding of the duration of IPT, 150 (38.4%) were aware of the recommended completion period, while 107 (27.4%)

**Table 1. Socio-demographic characteristics of respondents (n = 391).**

| Variable | Frequency | Percentage |
|---|---|---|
| **Sex** | | |
| Male | 112 | 28.6 |
| Female | 279 | 71.4 |
| **Age group (years)** | | |
| 18–31 | 65 | 16.6 |
| 32–45 | 154 | 39.4 |
| 46–59 | 121 | 30.9 |
| ≥60 | 51 | 13 |
| **Mean (SD)** | 44.0 | 12.7 |
| **Marital status** | | |
| Married | 186 | 47.6 |
| Single | 77 | 19.7 |
| Divorced/widowed/separated | 128 | 32.7 |
| **Level of education** | | |
| No formal education | 59 | 15.1 |
| Primary education | 282 | 72.1 |
| Secondary education | 41 | 10.5 |
| Certificate | 5 | 1.3 |
| Diploma/University | 4 | 1.0 |
| **Facility level** | | |
| Hospital | 97 | 24.8 |
| Health Centre | 183 | 46.8 |
| Dispensary | 111 | 28.4 |
| **Facility ownership** | | |
| Public | 323 | 82.6 |
| Faith-based | 51 | 4.3 |
| Private | 17 | 4.3 |
| **Occupation** | | |
| Unemployed | 19 | 4.9 |
| employed | 16 | 4.1 |
| Farmers | 235 | 60.1 |
| Business | 74 | 18.9 |
| **Location** | | |
| Urban | 132 | 33.8 |
| Rural | 259 | 66.2 |
| **Duration in HIV care** | | |
| Less than one year | 52 | 13.3 |
| One year and above | 339 | 86.7 |

were certain. When asked about the target groups for IPT services, 114 (29.5%) did not know them, while 32 (8.2%) expressed absolute certainty (**Table 2**).

## Relationship between participants' demographic characteristics and IPT knowledge level

A bivariate analysis was conducted to examine the relationship between participants' knowledge levels and various demographic characteristics. Of the 391 participants, 279 (71.4%) demonstrated adequate knowledge. In terms of sex, a statistically significant association was

**Table 2. Responses on IPT knowledge questions among participants (n = 391).**

| Variable | Frequency | Percentage (%) |
|---|---|---|
| **Do you think IPT can prevent you from TB?** | | |
| Absolutely No | 17 | 4.35 |
| No | 5 | 1.3 |
| Not sure | 60 | 15.4 |
| Yes | 188 | 48.1 |
| Absolutely Yes | 121 | 31 |
| **Do you know where to get IPT services?** | | |
| Absolutely No | 10 | 2.6 |
| No | 26 | 6.7 |
| Not sure | 24 | 6.1 |
| Yes | 196 | 50.1 |
| Absolutely Yes | 135 | 34.5 |
| **Do you know how long the IPT dose takes?** | | |
| Absolutely No | 20 | 5.1 |
| No | 58 | 14.8 |
| Not sure | 56 | 14.3 |
| Yes | 150 | 38.4 |
| Absolutely Yes | 107 | 27.4 |
| **Do you know groups of people are targeted for IPT?** | | |
| Absolutely No | 37 | 9.5 |
| No | 114 | 29.5 |
| Not sure | 109 | 27.9 |
| Yes | 99 | 25.3 |
| Absolutely Yes | 32 | 8.2 |

observed ($p<0.001$), with 185 (78.4%) of females exhibiting adequate knowledge compared to 94 (60.6%) of males. Age groups did not show a significant association with knowledge ($p = 0.348$). Marital status did not show any significant association ($p = 0.108$), although divorced/widowed/separated individuals showed a higher percentage of adequate knowledge 100 (78.1%) compared to singles 51 (66.2%) and married participants 128 (68.8%). Residence was associated with knowledge $p = 0.001$, with urban facilities 177 (77.6%) having adequate knowledge compared to rural facilities 102 (62.6%). Duration in treatment and care center was also associated with knowledge $p = 0.019$ (**Table 3**).

## Determinants of knowledge of IPT among PLHIV

The multivariable logistic regression analysis was used to examine the association between knowledge of TB prevention and various demographic characteristics among the participants. The adjusted odds ratios (AOR) and 95% confidence intervals (CI) were calculated for each variable. Female participants demonstrated a significantly higher likelihood of adequate knowledge compared to their male counterparts (AOR = 2.64, 95% CI: 1.36–5.15, p = 0.004). In terms of education, secondary education (AOR = 1.75, 95% CI: 1.00–3.08, p = 0.050) was associated with higher knowledge levels compared to those with no formal education. Participants from the urban location were two times (AOR = 2.28, 95% CI: 1.95–2.67, $p<0.001$) likely to have adequate knowledge as compared to participants from rural. Additionally, participants with one year and above in HIV care had higher odds of adequate knowledge (AOR = 2.24, 95% CI: 2.02–2.48, p < 0.001). The other variables, such as facility level, facility

**Table 3. Bivariate analysis of the relationship between knowledge and demographic characteristics of participants (n = 391).**

| Variable | Adequate knowledge n (%) | Inadequate knowledge n (%) | p-value |
|---|---|---|---|
| | 279(71.4) | 112(28.6) | |
| **Sex** | | | |
| Male | 94(60.6) | 61(39.4) | <0.001 |
| Female | 185(78.4) | 51(21.6) | |
| **Age group (years)** | | | |
| 18–31 | 43(66.2) | 22(33.8) | 0.348 |
| 32–45 | 111(72.1) | 43(27.9) | |
| 46–59 | 92(76.0) | 29(24.0) | |
| ≥60 | 33(64.7) | 18(35.3) | |
| **Marital status** | | | |
| Single | 51(66.2) | 26(33.8) | 0.108 |
| Married | 128(68.8) | 58(31.2) | |
| Divorced/widowed/separated | 100(78.1) | 28(21.9) | |
| **Level of education** | | | |
| No formal education | 41(66.1) | 21(33.9) | 0.553 |
| Primary education | 206(72.8) | 77(27.2) | |
| Secondary education | 32(69.6) | 14(30.4) | |
| **Facility level** | | | |
| Hospital | 70(72.2) | 27(27.8) | 0.951 |
| Health Centre | 131(71.6) | 52(28.4) | |
| Dispensary | 78(70.3) | 33(29.7) | |
| **Facility ownership** | | | |
| Public | 236(73.1) | 87(26.9) | 0.238 |
| Faith-based | 10(58.8) | 7(41.2) | |
| Private | 33(64.7) | 18(35.3) | |
| **Occupation** | | | |
| Unemployed | 24(68.6) | 11(31.4) | 0.803 |
| employed | 25(69.4) | 11(30.6) | |
| Farmers | 168(70.6) | 70(29.4) | |
| Business | 62(75.6) | 20(24.4) | |
| **Location** | | | |
| Rural | 102(62.6) | 61(37.4) | 0.001 |
| Urban | 177(77.6) | 51(22.4) | |
| **Distance from home to health facility** | | | |
| Less than 2km | 65(27.7) | 170(72.3) | 0.597 |
| 2km and above | 47(30.1) | 109(69.9) | |
| **Duration in care and treatment center** | | | |
| Less than one year | 30(57.7) | 22(42.3) | 0.019 |
| One year and above | 249(73.4) | 90(26.6) | |

ownership, occupation, and distance from home, did not show significant associations with knowledge levels (**Table 4**).

## Discussion

The findings of this study show that more than a third of PLHIV demonstrated an overall adequate knowledge of IPT. These findings differed from a similar study conducted in South East Nigeria [8] which found an overall lower knowledge level of 39.5%. The variance in knowledge

**Table 4. A multivariable logistic regression analysis showing the relationship between knowledge and demographic characteristics.**

| Variable | AOR (95%CI) | *p*-value |
|---|---|---|
| **Sex** | | |
| Male | 1 | |
| Female | 2.64(1.36–5.15) | 0.004 |
| **Age group (years)** | | |
| 18–31 | 1 | |
| 32–45 | 0.85(0.48–1.50) | 0.572 |
| 46 years and above | 0.91(0.34–2.44) | 0.851 |
| **Marital status** | | |
| Single | 1 | |
| Married | 1.85(0.42–8.15) | 0.416 |
| Divorced/widowed/separated | 2.30(2.15–2.45) | <0.001 |
| **Level of education** | | |
| No formal education | 1 | |
| Primary education | 1.56(0.92–2.66) | 0.101 |
| Secondary education | 1.75(1.00–3.08) | 0.050 |
| **Facility level** | | |
| Hospital | 1 | |
| Health Centre | 0.83(0.40–1.74) | 0.630 |
| Dispensary | 0.93(0.63–1.35) | 0.688 |
| **Facility ownership** | | |
| Public | 1 | |
| Private | 0.41(0.17–1.01) | 0.053 |
| Faith-based | 0.60(0.14–2.61) | 0.492 |
| **Occupation** | | |
| Unemployed | 1 | |
| employed | 1.19(0.36–3.95) | 0.782 |
| Farmers | 1.13(0.60–2.11) | 0.706 |
| Business | 1.14(0.85–1.53) | 0.375 |
| **Location** | | |
| Rural | 1 | |
| Urban | 2.28(1.95–2.67) | <0.001 |
| **Distance from home to health facility** | | |
| Less than 2km | 1 | |
| 2km and above | 0.68(0.56–0.83) | <0.001 |
| **Duration in care and treatment centers** | | |
| Less than one year | 1 | |
| One year and above | 2.24(2.02–2.48) | <0.001 |

levels may be attributed to temporal factors suggesting potential shifts in awareness over time because the Nigerian study was conducted in 2018 while the current study was conducted in 2023. Moreover, the geographical as well as social and political differences in the study populations may account for the recorded difference as the former study was conducted in Western parts of Africa while this one comes from the Eastern part.

The observed level of TPT knowledge in the study implies success in adapting the WHO recommendation for enhancing knowledge of TPT among PLHIV as a key intervention to reduce the socioeconomic burden of active TB on PLHIV and the general population [11].

Regarding the determinants of knowledge, our study found that being female, attending an urban-based health facility for IPT services, and being on HIV care for more than a year were important factors influencing knowledge of IPT among PLHIV.

A significant association between gender and TPT knowledge was observed. Specifically, female participants were identified as being more than two times more likely to have adequate TPT knowledge compared to their male counterparts. These findings conform with the study conducted in Nigeria [8] and Ethiopia [12] which all together reported that female participants had adequate IPT knowledge compared to male participants. These findings suggest that there may be gender-specific factors, such as differences in healthcare-seeking behavior, awareness, or health education exposure influencing the level of knowledge among PLHIV about TPT. Understanding these gender-related disparities is crucial for the development of targeted interventions to enhance TPT knowledge, ensuring that both male and female populations receive adequate information and education about preventive therapy for TB.

The current study revealed that the location of a health facility was significantly associated with the level of knowledge on IPT among PLHIV. We found that PLHIV who received IPT care from urban facilities were more than twice as likely to have sufficient IPT knowledge compared to those receiving care in rural health facilities. These findings suggest that there may be factors related to the urban setting, such as better access to information, communication channels, and educational resources [9] which may contribute to a higher level of IPT knowledge among PLHIV from urban locations. The NTPs and other TB control stakeholders have a critical task in implementing targeted interventions on awareness campaigns in rural areas. These locations require such interventions the most to address the urban-rural knowledge disparities among PLHIV particularly on TB preventive treatment.

The study revealed that duration of care was an important factor for knowledge of IPT among PLHIV. We found that participants who had longer than a year of HIV care had an adequate level of knowledge of IPT compared to those who had just joined HIV care. These findings concur with the findings of a study conducted in Ethiopia which found that PLHIV who were on care for longer than six months were more than two times likely to be informed about IPT compared to those who had less than six months on care [12]. It is important to note that this association could be attributed to several factors including longer durations of interactions with healthcare providers and participation in health education sessions during HIV treatment. Additionally, individuals who have been in HIV care for an extended duration may have had more exposure to informational materials, counseling, and awareness campaigns, leading to a better understanding of IPT.

## Conclusion

While a substantial portion of PLHIV demonstrated an overall adequate knowledge of IPT, the significance of addressing IPT knowledge gaps among PLHIV who were found to have low levels of knowledge cannot be underestimated. These findings underscore the importance of ongoing education and awareness campaigns to ensure that PLHIV has accurate and comprehensive knowledge about TPT. Additionally, the study highlights the significance of demographic factors such as gender, location of health facilities, and duration of HIV care, which influence IPT knowledge among PLHIV. Addressing these determinants through targeted interventions can contribute to improving health outcomes and ultimately reducing the burden of TB among PLHIV and the general population. It is crucial to foster collaboration between the NTLP, The National AIDS, STIs, and Hepatitis Control Program (NASHCoP), and community organizations to integrate TPT awareness interventions into comprehensive HIV care programs, enhancing their reach and effectiveness among PLHIV.

## Study limitations

The study regions were selected based on the funder's area of support. This sampling may limit the country-wide generalizability of the results of this study. Despite our efforts, logistical challenges during data collection hindered us from reaching the desired sample size within the allocated timeframe.

## Acknowledgments

We sincerely acknowledge the support from the District TB coordinators from study regions and the cooperation we received from data clacks at Care and Treatment Clinics whose contribution significantly impacted the success of this work.

## Author Contributions

**Conceptualization:** Felix Christopher Alexander, Johnson Jeremia Mshiu, Anelisa Martin Rushaigo, Victor Enock Wiketye, Segere Chacha Mtundi, Sylvia Thomas Haule, Michael Kipenda Katende, Esther Manka Evarist, Luciana Charles Kapama, Aloisia Ibrahim Shemdoe, Charles Elias Makasi, Majaha Melkisedeck Lolakeha, Victor Kenedy Minja, Omary Abdallah Kimbute, Kunda John Stephen, Nyagosya Segere Range, Werner Meinrad Maokola, Bernard James Ngowi, Vitus Alberto Nyigo, Andrew Martin Kilale.

**Data curation:** Felix Christopher Alexander, Erick Josephat Mgina, Andrew Martin Kilale.

**Formal analysis:** Felix Christopher Alexander, Johnson Jeremia Mshiu, Erick Josephat Mgina, Andrew Martin Kilale.

**Funding acquisition:** Andrew Martin Kilale.

**Investigation:** Felix Christopher Alexander, Johnson Jeremia Mshiu, Andrew Martin Kilale.

**Methodology:** Felix Christopher Alexander, Andrew Martin Kilale.

**Project administration:** Andrew Martin Kilale.

**Resources:** Andrew Martin Kilale.

**Supervision:** Kunda John Stephen, Nyagosya Segere Range, Werner Meinrad Maokola, Bernard James Ngowi, Vitus Alberto Nyigo, Andrew Martin Kilale.

**Validation:** Felix Christopher Alexander, Johnson Jeremia Mshiu, Kunda John Stephen, Nyagosya Segere Range, Werner Meinrad Maokola, Bernard James Ngowi, Vitus Alberto Nyigo.

**Visualization:** Felix Christopher Alexander, Johnson Jeremia Mshiu, Andrew Martin Kilale.

**Writing – original draft:** Felix Christopher Alexander, Johnson Jeremia Mshiu.

**Writing – review & editing:** Felix Christopher Alexander, Johnson Jeremia Mshiu, Anelisa Martin Rushaigo, Erick Josephat Mgina, Victor Enock Wiketye, Segere Chacha Mtundi, Sylvia Thomas Haule, Michael Kipenda Katende, Esther Manka Evarist, Luciana Charles Kapama, Aloisia Ibrahim Shemdoe, Charles Elias Makasi, Majaha Melkisedeck Lolakeha, Victor Kenedy Minja, Omary Abdallah Kimbute, Kunda John Stephen, Nyagosya Segere Range, Werner Meinrad Maokola, Bernard James Ngowi, Vitus Alberto Nyigo, Andrew Martin Kilale.

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
