## [Decision Letter · Decision Letter 0]

13 May 2024

PONE-D-24-10620Knowledge of Tuberculosis Preventive Treatment among People Living with HIV: A cross-sectional survey in selected regions of TanzaniaPLOS ONE

Dear Dr. Alexander,

Thank you for submitting your manuscript to PLOS ONE. After careful consideration, we feel that it has merit but does not fully meet PLOS ONE’s publication criteria as it currently stands. Therefore, we invite you to submit a revised version of the manuscript that addresses the points raised during the review process.

https://journals.plos.org/plosone/s/file?id=ba62/PLOSOne_formatting_sample_title_authors_affiliations.pdf"

"This study received funding support from the Global Fund through NASHCoP of the Ministry of Health 

Tanzani"

"Our study received funding support from the Global Fund to fight against HIV, TB and Malaria in Tanzania through the NASHCoP. However, the funder did not take part in the study design, data collection and analysis, decision to publish, nor preparation of this manuscript.

AMK received the fund. 

 https://www.theglobalfund.org/en/results/ "

Please include your amended statements within your cover letter; we will change the online submission form on your behalf."""

Additional Editor Comments:

The authors have presented a paper evaluating the knowledge of TPT among PLWHIV in Tanzania.

Major comments

1. The calculated sample size for this study is 427 and the number of participants enrolled was 391. Not meeting the sample size means the results lack statistical power.

2. Another important factor to consider when evaluating knowledge about TPT is, "Time since enrolling into care" It is likely that those who have been in care longer and those who have actually received the TPT will be more knowledgeable. This variable needs to be evaluated.

3. There needs to be stronger justification by the authors why they think that PLWHIV should be knowledgeable about TPT. Patients do not prescribe this modality to themselves so knowledge about TPT is the purview of healthcare workers and not individuals that have received a diagnosis of HIV.

Minor comments

1. Consider having the paper reviewed for grammar. There are some grammatical errors that need to be corrected e.g. line 67, "Daily" should be, "daily" line 87, "Prevention" should be, "prevention"

2. Line 113: Define the abbreviation CTC in the paper before using it.

3. IPT and TPT cannot be used interchangeably. This paper evaluated IPT knowledge. IPT is a strategy for TPT. So, the title is not precise because it refers to TPT while the study evaluated IPT. Therefore even in the paper, the use of the acronym TPT has to be precise: line 117 and page 15 of the paper.

4. In line 142 please state who (9) and (10) are then reference them.

Reviewers' comments:

**Comments to the Author**

Reviewer #1: An interesting and, in the context of the authors' country, an important study has been conducted.

I have a few major questions that I would like to clarify:

• The manuscript lacks context about the TB and HIV situation in Tanzania. In lines 65-66 it is written "In 2022, the proportion of people with TB coinfected with HIV stood at 17%". Regarding this study, much more important information would be what proportion of HIV-infected persons have (had) active TB and what proportion of HIV-infected persons have latent TB infection?

• The authors do not mention anywhere the prevalence of latent TB infection and its diagnosis in Tanzania. Preventive treatment is given in the case of latent TB infection. Clarity is needed.

• In lines 66-69, the statement "The National TB and Leprosy Program (NTLP) adopted the WHO recommendations for TB prevention among high-risk populations thus recommending daily Isoniazid as the main preventive drug for PLHIV and under-five household contacts of people with active TB "unclear. Different situations are discussed in one sentence - HIV infection (with latent TB infection) and contact persons with a patient with active TB. Need to clarify.

Minor concerns:

• In line 70, the abbreviation ISP is not expanded.

• In line 14 is probably misspelled "Tanzani".

Reviewer #2: This study examines knowledge of TB preventive treatment among a sample of people living with HIV in Tanzania. Overall, the study was very well written and easy to understand. The study findings were presented in a clear way and all tables were well structured and easy to interpret. The study discusses an important topic in HIV prevention, and the authors give plausible explanations of the key determinants of TPT knowledge that their study identified (e.g., gender, location etc). I have no additional comments. The paper was well written and it successfully addressed its central questions.

We look forward to receiving your revised manuscript.

Kind regards,

Tinashe Mudzviti, MPhil(MD)

Academic Editor

PLOS ONE

---

## [Author Response · Author response to Decision Letter 0]

23 May 2024

'When completing the data availability statement of the submission form, you indicated that you will make your data available on acceptance. We strongly recommend all authors decide on a data sharing plan before acceptance, as the process can be lengthy and hold up publication timelines. Please note that, though access restrictions are acceptable now, your entire data will need to be made freely accessible if your manuscript is accepted for publication. This policy applies to all data except where public deposition would breach compliance with the protocol approved by your research ethics board. If you are unable to adhere to our open data policy, please kindly revise your statement to explain your reasoning and we will seek the editor's input on an exemption. Please be assured that, once you have provided your new statement, the assessment of your exemption will not hold up the peer review process'

I received this from the editor, but I had from the previous submission made the data available in Frig share and attached the Doi: . The data freely accessible and I ticked

---

## [Editor Report · Decision Letter 1]

19 Jun 2024

PONE-D-24-10620R1Knowledge of Tuberculosis preventive treatment among people living with HIV: A cross-sectional survey in selected regions of TanzaniaPLOS ONE

Dear Dr. Alexander,

Thank you for submitting your manuscript to PLOS ONE. After careful consideration, we feel that it has merit but does not fully meet PLOS ONE’s publication criteria as it currently stands. Therefore, we invite you to submit a revised version of the manuscript that addresses the points raised during the review process.

**ACADEMIC EDITOR: ** Authors need to include the limitation in the manuscript that there is a reduction in the power of the results which was a result of not meeting the sample size.

We look forward to receiving your revised manuscript.

Kind regards,

Tinashe Mudzviti, MPhil(MD)

Academic Editor

PLOS ONE
---

## [Author Response · Author response to Decision Letter 1]

26 Jun 2024

I have attached my response on the study limitations which made us fail to reach the sample size.

---

## [Editor Report · Decision Letter 2]

10 Jul 2024

Knowledge of Tuberculosis preventive treatment among people living with HIV: A cross-sectional survey in selected regions of Tanzania

PONE-D-24-10620R2

Dear Dr. Alexander,

We’re pleased to inform you that your manuscript has been judged scientifically suitable for publication and will be formally accepted for publication once it meets all outstanding technical requirements.

Kind regards,

Tinashe Mudzviti, MPhil(MD)

Academic Editor

PLOS ONE
---

## [Editor Report · Acceptance letter]

24 Jul 2024

PONE-D-24-10620R2 

PLOS ONE

Dear Dr. Alexander, 

I'm pleased to inform you that your manuscript has been deemed suitable for publication in PLOS ONE. Congratulations! Your manuscript is now being handed over to our production team.

Kind regards, 

on behalf of

Dr. Tinashe Mudzviti 

Academic Editor

PLOS ONE